genetics/health and disease and epidemiology/
physiology

*Bombyx mori*, *lemon* mutant, *BmSPR*,
sepiapterin reductase deficiency,
animal model of human diseases

**Author for correspondence:**
Fangyin Dai
e-mail: fydai@swu.edu.cn

# Evaluation of the silkworm *lemon* mutant as an invertebrate animal model for human sepiapterin reductase deficiency

Guihua Jiang, Jiangbo Song, Hai Hu, Xiaoling Tong
and Fangyin Dai

State Key Laboratory of Silkworm Genome Biology, Key Laboratory of Sericulture Biology and
Genetic Breeding, Ministry of Agriculture and Rural Affairs, College of Biotechnology,
Southwest University, Chongqing 400715, People's Republic of China

FD, 0000-0002-0215-2177

Human sepiapterin reductase (SR) deficiency is an inherited disease caused by *SPR* gene mutations and is a monoamine neurotransmitter disorder. Here, we investigated whether the silkworm *lemon* mutant could serve as a model of SR deficiency. A point mutation in the *BmSPR* gene led to a five amino acid deletion at the carboxyl terminus in the *lemon* mutant. In addition, classical phenotypes seen in SR deficient patients were observed in the *lemon* mutant, including a normal phenylalanine level, a decreased dopamine and serotonin content, and an increased neopterin level. A recovery test showed that the replenishment of L-dopa significantly increased the dopamine level in the *lemon* mutant. The silkworm *lemon* mutant also showed negative behavioural abilities. These results suggest that the silkworm *lemon* mutant has an appropriate genetic basis and meets the biochemical requirements to be a model of SR deficiency. Thus, the silkworm *lemon* mutant can serve as a candidate animal model of SR deficiency, which may be helpful in facilitating accurate diagnosis and effective treatment options of SR deficiency.

## 1. Introduction

Many methods are used to study various diseases, such as mathematical models, cell models and patient volunteers. However, each of these methods has limitations. For example, computer models are limited by known disease information. Additionally, cell studies do not reflect the true systemic situation

of the individual, because diseases reflect the combined action of genes and environment. Often, it is too dangerous to carry out studies in patients directly and doing so involves ethical issues. To solve these problems, animal models of human disease are used in the laboratory. Animal models of human disease have many advantages. For example, there is no spatio-temporal limit. It is easy to research human diseases that have long duration and low incidence. Researchers also can study the effects of various conditions on the disease. To date, there are more than 5000 animal models of human diseases. Common species used as animal models of human diseases include monkey (Primates) [1], mouse (*Mus musculus*) [2], zebrafish (*Danio rerio*) [3], fruit fly (*Drosophila melanogaster*) [4] and nematode (*Caenorhabditis elegans*) [5]. The main diseases studied include acquired immune deficiency syndrome, cancer, diabetes, nervous system diseases and cardiovascular diseases. These animal models have been chosen for their respective advantages and characteristics. However, they still have shortcomings, such as limited resources, expensive experimental costs, long experimental periods and animal ethics and welfare disputes. Silkworms (*Bombyx mori*) are not only economic insects, but have also been useful model animals in genetics for more than a century. Additionally, the silkworm has 8469 human homologous genes with 58% homology [6].

The silkworm has many advantages as an animal model of human disease, such as abundant mutant resources, a clear genetic background, a short life cycle, a high fertility rate, a moderate body size, convenient dissection, easy operation for haemolymph and midgut injection, as well as no animal ethical controversy. Until now, there have been some reports on the use of silkworms to construct disease models, such as a hyperlipidemic model [7], a hyperproteinemic model and a hyperglycemic model [8,9]. In addition, there are silkworm mutants that have the potential to be used as the models of human diseases. For example, the silkworm *oa* mutant is caused by a mutation in the *BmHPS5* gene, which is homologous with the Hermansky–Pudluck syndrome-5 (*HPS5*) gene in humans [10]. Down-regulation of the *DJ-I* gene in the silkworm *op* mutant affects the plasma uric acid synthesis-modulating pathway and results in a similar phenotype to the clinical features of Parkinson disease [11]. The pale body colour silkworm (*albino, al*) is caused by mutation of the *BmPTPS* gene, and the results of related gene expression analysis and recovery experiments indicated that the mutant might be a potential animal model of tetrahydrobiopterin (BH4)-deficient phenylketonuria [12,13]. The silkworm also plays a role in drug screening, and there have been pathogenic bacterial infection models [14], pathogenic fungal infection models [15], as well as antibiotic drug screening models [16]. Therefore, current studies provide an important theoretical basis for using the silkworm to analyse the pathogenesis of human diseases and develop therapeutic drugs.

BH4 has vital functions as a cofactor of multiple enzymes, including phenylalanine hydroxylase (PAH), tryptophan hydroxylase (TPH), tyrosine hydroxylase (TH) and nitric-oxide synthase (NOS) [17,18]. BH4 also participates in other biological processes, such as erythroid cell proliferation [19], human melanogenesis [20] and cell-mediated immunity [21]. In mammals, there are three biosynthetic pathways for BH4 (figure 1) [22]. The de novo biosynthetic pathway of BH4 begins from guanosine triphosphate (GTP) via three catalysed reactions by GTP cyclohydrolase I (GTPCH I), 6-pyruvoyl-tetrahydropterin synthase (PTPS) and sepiapterin reductase (SR) [22]. GTPCH I is a rate-limiting factor in the pathway, and NADPH is indispensable. In particular, when the pathway is dysregulated, 6-pyruvoyl-tetrahydropterin can be transformed into 1′-oxo-PH4 by aldose reductase (AR) and carbonyl reductase (CR) [23]. CR converts the sepiapterin derived from 1′-oxo-PH4 non-enzymatically into 7, 8-dihydrobiopterin (BH2), which is reduced to BH4 by dihydrofolate reductase. This is the second source for BH4, the salvage pathway. In the regeneration pathway of this cofactor, pterin-4α-carbinolamine becomes BH4 by a series of actions requiring pterin-4α-carbinolamine dehydroxylase (PCD) and dihydropteridine reductase (DHPR) [22]. When key enzymes in the above three pathways are dysregulated, BH4 homeostasis is disturbed, leading to a series of serious problems. For instance, a deficiency of BH4 may result in monoamine neurotransmitter disorders [24], because the generation of dopamine, serotonin and other neurotransmitters are limited by TH and TPH, which cannot function normally without BH4 [25]. There are four types of BH4 deficient-monoamine neurotransmitter disorders, including GTPCH deficiency (Segawa's disease, OMIM: 128230) [26], PTPS deficiency (OMIM: 261640) [27], DHPR deficiency (OMIM: 261630) [28] and PCD deficiency (OMIM: 126090) [29].

In 2001, SR deficiency (OMIM: 612716), another type of BH4 deficiency without hyperphenylalaninemia, was fully recognized [30]. SR deficiency is an inherited autosomal recessive disorder [31], and its incidence is unknown. To date, 13 different mutations have been found in 44 SR deficient patients (http://www.biopku.org) [32]. SR deficiency is a neurotransmitter disorder caused by *SPR* gene (2p14-p12) mutations [31]. Its core clinical features mainly include cognitive problems, such as

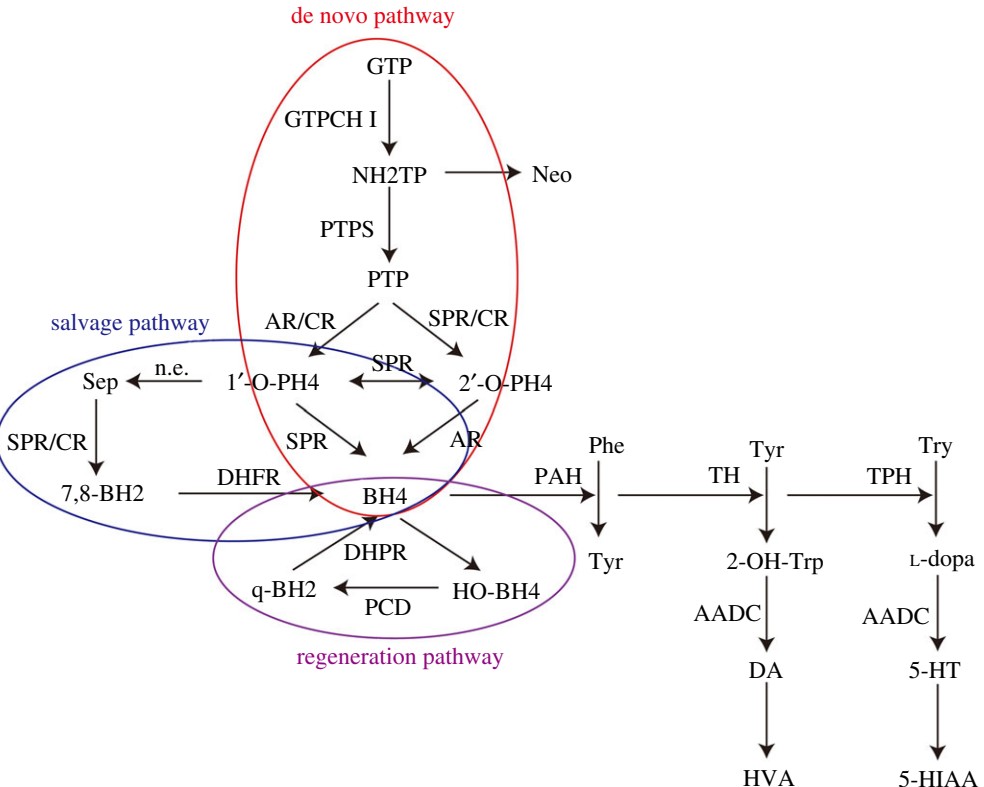

**Figure 1.** The biosynthesis and metabolic pathway of BH4. BH4 is a cofactor of PAH, TH and TPH and participates in the synthesis of neurotransmitters. GTPCH I, GTP cyclohydrolase; PTPS, PTP synthase; SR, sepiapterin reductase; CR, carbonyl reductase; AR, aldose reductase; DHFR, dihydrofolate reductase; PCD, pterin-4α-carbinolamine dehydratase; DHPR, dihydrobiopterin reductase; TH, tyrosine hydroxylase; TPH, tryptophan hydroxylase; PAH, phenylalanine hydroxylase; AADC, aromatic ʟ-amino acid decarboxylase.

mental retardation, extreme mood swings and language delay [33], and motor problems including dystonia, axial hypotonia, muscle stiffness and tremors, seizures and oculogyric crises [34,35]. SR deficiency cannot be detected by newborn screening because it is not companied by hyperphenylalaninemia [36] and initially is often misdiagnosed as other diseases, such as DHPR deficiency [37]. This results in physical damage owing to delayed treatment, which should be initiated as early as possible. The exact diagnosis must be made with mutation analysis of the *SPR* gene combined with an analysis of pterins and biogenic amines in cerebrospinal fluid [38]. SR deficiency displays decreased levels of homovanillic acid (HVA) and 5-hyroxyindolacetic acid (5-HIAA), two products of neurotransmitters, and elevated total biopterin (tBio), dihydrobiopterin (BH2) and sepiapterin, as well as normal to slightly increased levels of neopterin. The treatment of SR deficiency aims to correct central nervous system (CNS) dopamine deficiency, so the common method is to supply ʟ-dopa, the precursor of dopamine, and carbidopa, a peripheral decarboxylase inhibitor [39]. It is important to initiate treatment as early as possible to avoid irreversible neurological damage. Some challenges in SR deficiency remain. One is that newborn screening is difficult to obtain, but is key to prevent mental disability. Another problem is that the treatment method is too specific to meet the needs of all patients with different mutations, which brings great difficulties for clinical discovery and treatment of the disease. Thus, suitable animal models are needed to resolve these problems and to gain a better understanding of other similar human diseases.

In this study, we explored whether the *lemon* mutant could serve as an animal model of SR deficiency. Molecular cloning showed that the *BmSPR* gene of the *lemon* mutant had a point mutation. In addition, biochemical studies revealed a decreased level of dopamine and serotonin, an increased level of neopterin and a normal level of phenylalanine (LPA). Furthermore, the level of dopamine increased significantly after feeding ʟ-dopa. Additionally, the *lemon* mutant showed negative behavioural abilities. Interestingly, unlike the results for human patients, the *lemon* mutant displayed a normal level of blood trehalose. Our findings indicate that the *lemon* mutant can serve as a candidate animal model of human SR deficiency, which will play an important role to uncover the pathogenic mechanism and for screening drugs for SR deficiency and other similar monoamine neurotransmitter diseases.

**Table 1.** The primers used for PCR and RT-qPCR.

| gene name | sense primer | antisense primer |
|---|---|---|
| **PCR primers** | | |
| *BmActin3* | 5′AACACCCCGTCCTGCTCACTG3′ | 5′GGGCGAGACGTGTGATTTCCT3′ |
| *BmSPR* | 5′CTTATCAGCGTACAGAGCCGA3′ | 5′ATACGAAGACCCGACGAACAC3′ |
| **RT-qPCR primers** | | |
| *sw22934* | 5′TTCGTACTGCTCTTCTCG3′ | 5′CAAAGTTGATAGCAATTCCCT3′ |
| *BmPAH* | 5′CCCTCATACGGTGCCGAACT3′ | 5′CATCAGCAGCGGGAAGACAT3′ |
| *BmTH* | 5′TTGATGCCCAAACACGC3′ | 5′TCGCAGGGTAAAGCCAGT3′ |
| *BmGTPCH* | 5′GGCTAACTCATACAGGCTTCTAC3′ | 5′CTTCTGGACTCCTCGCATC3′ |
| *BmPTPS* | 5′ATGTCTTCTTTACCTATTGTATC3′ | 5′GTCAGAATCTGAGGAAACTTC3′ |
| *BmSPR* | 5′ATGGCTATGTCGTCTAGCATC3′ | 5′TTCGTCATCGAAATAGTCGAC3′ |
| *BmDHFR* | 5′ATGTCTCGTACGCAACTGAATTTGA3′ | 5′TTATAATCTCTTGTAAATCCTATAG3′ |

# 2. Material and methods

## 2.1. Reagents and drugs

Dopamine hydrochloride, serotonin, phosphoric acid, sodium 1-octanesulfonate monohydrate, L-ascorbic acid and methanol were bought from Sigma-Aldrich (USA). Hydrochloric acid (HCl, 36%) was purchased from Chuandong Chemical Group (Chongqing, China). Sodium hydroxide (NaOH) was obtained from Chron Chemicals (Chengdu, China). Disodium edetate dehydrate was obtained from BBI Life Sciences. Dichloromethane was bought from Shanghai Macklin Biochemical (Shanghai, China).

## 2.2. Silkworm strain and feeding conditions

All silkworm strains used in the experiments were obtained from the Silkworm Gene Bank at Southwest University. Their environment was 25°C, 75% relative humidity, and a 12 L : 12 D photoperiod. Mulberry leaves were their only food during the entire larval stage [40].

## 2.3. Cloning of the *BmSPR* gene

We downloaded the full-length sequence of the BmSPR gene from SilkDB (silkworm.genomics.org.cn) and designed primer pairs to target the whole coding sequence (CDS). After sequencing cDNA clones, we aligned the obtained sequence with the sequence submitted on NCBI. The primers used for polymerase chain reaction (PCR) are shown in table 1.

## 2.4. Reverse transcription-quantitative polymerase chain reaction

The silkworms were collected from day 1 of the first instar to day 1 of the fifth instar. The Total RNA Kit (BioTeke Corporation, Beijing, China) was used to extract the total RNA of all samples based on the manufacturer's instructions. After using the PrimeScriptTM RT Reagent Kit with gDNA Eraser (TaKaRa, Dalian, China) to obtain cDNA from the extracted total RNA samples, reverse transcription-quantitative PCR (RT-qPCR) experiments were carried out by iTaq Universal SYBR Green Supermix and a CFX96 RealTime System (Bio-Rad, Shanghai, China). The eukaryotic translation initiation factor 4A (*BmMDB*, probe identity: *sw22934*) was chosen as the reference gene. The primer pairs shown in table 1 were designed to target each gene, including *sw22934*, *BmSPR*, *BmGTPCH I*, *BmPTPS*, *BmDHFR*, *BmPAH* and *BmTH*, and the experimental conditions followed the manufacturer's instructions [40,41].

## 2.5. Measurement of body weight and length

The wild-type and *lemon* mutants at day 3 of fifth instar larvae were selected randomly and split into three groups to measure the body length and weight before feeding [40].

## 2.6. Behavioural tests

We designed a mulberry leaf-luring test and a turnover test to investigate the locomotor activity of the *lemon* mutant. In the mulberry leaf-luring test, silkworm *lemon* mutants and wild-type silkworms were chosen randomly 5 h after feeding. The two groups of silkworms were placed 0.5 cm away from the chronograph line, which is the pre-crawl distance. At 5 cm away from the chronograph line, we put mulberry leaves to induce silkworm crawling. When the silkworms climbed to the timing line, we began to count. The number of silkworms crawling to the mulberry leaves and the time required by each group were counted within 15 min. In the turnover test, the amount of time required for the silkworm to turn upside down was measured. Day 3 of fifth instar larvae of silkworm *lemon* mutants and wild-type silkworms were chosen for the two tests. Scatter plots of the two tests were drawn using the Column mode in GraphPad Prism5.

## 2.7. Serum phenylalanine and trehalose assays

The serum phenylalanine level was measured by a kit purchased from Suzhou Comin Biotechnology Co. Ltd (Suzhou, China). The serum trehalose level was measured using a kit from Beijing Suo Laibao Biotechnology Co. Ltd (Beijing, China). All steps followed the manufacturer's instructions.

## 2.8. Neopterin measurements

The ultra performance liquid chromatograph (UPLC) system (Shimadzu, Japan) used here consists of a SIL-30AC automatic sampler, an LC-30AD solvent delivery module, a CTO-30A column oven, a CBM-20A system controller, an RF-20A fluorescence detector and an LC solution workstation. The silkworm brains were removed immediately and placed into 1.5 ml centrifuge tubes containing 50 mM Tris-HCl (pH 7.5), 0.1 M KCl, 1 mM EDTA, 1 mM dithiothreitol and proteinase inhibitors by use of a JXFSTPRP-32 grinding miller (Shanghai Jingxin Industrial Development Co. Ltd, China) [37]. About 1 ml of the homogenizing buffer was added to each sample. After homogenizing, the mixtures were centrifuged at 12 000 rpm at 4° C for 20 min. Then, the supernatant was filtered using a 0.22 µm filter (Tianjin Jinteng Experiment Co. Ltd, China) and stored at −80°C. A T3 C18 column (1.8 µm, 2.1 internal diameter ×100 mm) (Acquity UPLC HSS) was used to separate the chemicals under the mobile phase isopropanol : methanol : acetic acid : $H_2O$ (0.5:0.5:0.5:98.5, v/v, pH 2.68) with a flow rate of 0.15 ml min$^{-1}$. Neopterin was detected at $\lambda_{ex/em}$ 350/450 nm [37,42].

## 2.9. Dopamine and serotonin measurements

The silkworm brains were removed into a cold homogenizing buffer and blended using a JXFSTPRP-32 homogenizer for 240 s at 70 Hz. The homogenizing buffer contained 444.5 mg L-ascorbic acid and 61 ml hydrochloric acid (36%) with water to 500 ml. Afterward, the sample was centrifuged at 12 000 rpm for 20 min at 4°C. The supernatants were transferred into clean 1.5 ml microcentrifuge tubes in a boiling water bath for 3 min. After cooling, the same volume of dichloromethane was added to the supernatants and vortexed (PTR-60) for 10 min. After centrifuging the mixture at 12 000 rpm for 15 min, the supernatants were frozen at −80°C until use. We used a Shim-pack XR-ODS III column (1.6 µm, 2.0 mm internal diameter × 75 mm). The mobile phase was a mixture of 0.021 g 1-octanesulfonate monohydrate, 0.038 g disodium edetate dehydrate, 6.78 ml phosphoric acid and 100 ml methylalcohol. After adjusting to pH 2.68 by NaOH, the mobile phase was filtered using 0.22 µm filters. The separation was performed with a flow rate of 0.2 ml min$^{-1}$ at 40°C. Dopamine and serotonin were detected at $\lambda_{ex/em}$ 280/330 nm. The UPLC system was the same as that used for neopterin measurements.

## 2.10. L-dopa feeding experiment

Each *lemon* mutant was fed with mulberry leaves with 4 mM, 6 mM and 8 mM L-dopa and carbidopa mixture (treatment group) or deionized water (control group) from day 1 to day 3 of fifth instar larvae. The dopamine level in the brains of the four groups was detected by UPLC.

## 2.11. Statistical analysis

GraphPad Prism 5 software (La Jolla, CA, USA) was used to analyse the statistical data. The results are shown as mean ± s.e.m. and the significance was determined with a two-tailed Student's *t*-test. Values of $p < 0.05$ were statistically significant.

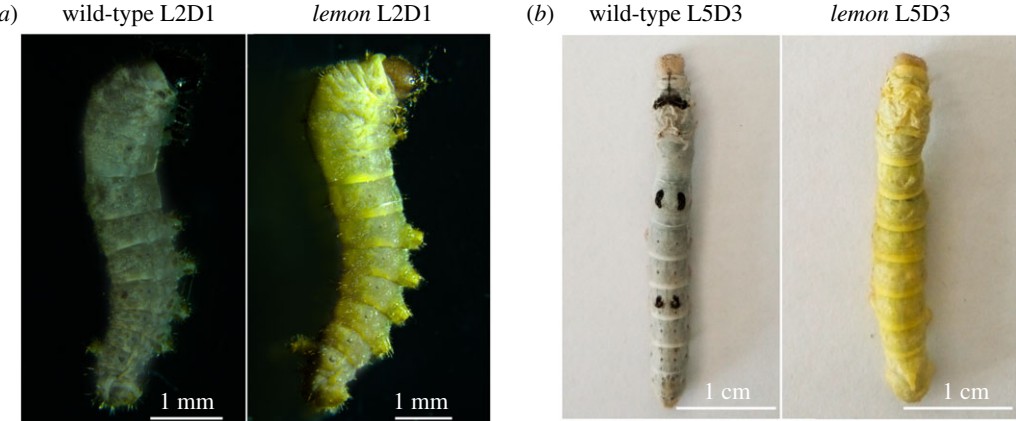

**Figure 2.** A comparison of phenotypes between the *lemon* mutant and the wild-type: (*a*) day 1 of second larval instar; (*b*) day 3 of fifth larval instar.

# 3. Results

## 3.1. Yellow body colour and *BmSPR* gene mutation in the silkworm *lemon* mutant

Silkworm *lemon* mutant, a spontaneous autosomal recessive mutant, has yellow body colour like the Japanese *lem* mutant [43], in contrast with the normal pigmentation of the wild-type (figure 2), which might be caused by abnormalities in the pigment metabolism pathway. Body colour pigments in insects mainly include ommochrome, pterin and melanin. Previous studies have reported that the yellow body coloration of the Japanese *lem* mutant is because of the accumulation of sepiapterin, yellow pteridines and sepialumazine in the integument [44]. In 2009, Meng *et al.* [43] demonstrated that the *BmSPR* gene was responsible for the Japanese *lem* mutant. Therefore, we investigated mutations in the *BmSPR* gene in the *lemon* mutant. We found that the *BmSPR* gene of the *lemon* mutant had a point mutation same to the change of the gene in the Japanese *lem* mutant (figure 3*a*), which leads to premature translation termination and a five amino acid deletion at the carboxyl terminus (figure 3*b*). Furthermore, we investigated the relationship between this yellow body colour and the *BmSPR* gene of the *lemon* mutant through a multi-line validation experiment. We found that the *BmSPR* gene of the nine yellow body colour strains had the same point mutation (figure 3*c,d*), indicating that the yellow body colour is related to the *BmSPR* gene. This phenotype suggests that the silkworm *lemon* mutant may be an appropriate animal model. These data showed that the *lemon* mutant has the genetic basis as an animal model of human SR deficiency, which is an inherited disease caused by *SPR* gene mutations.

## 3.2. Higher expression of the *BmSPR* gene at early larva instar in the silkworm *lemon* mutant

To understand the expression of the *BmSPR* gene in the *lemon* mutant, we investigated the temporal expression profile of *BmSPR* in the wild-type and *lemon* mutant. The results showed that the *BmSPR* gene is expressed in each period (from first instar to fifth instar) (figure 4*a*). Among them, the expression level of the *BmSPR* gene in the *lemon* mutant was higher during the first day of the second instar than in other periods, which indicated that the *BmSPR* gene is more important in the early stage of silkworm larvae. This is consistent with human SR deficiency, which shows clinical features in infancy. We also investigated the expression level of important genes in the $BH_4$ synthesis pathway, including *BmGTPCH I*, *BmPTPS*, *BmDHFR*, *BmPAH* and *BmTH* in the *lemon* mutant during the early stages (figure 4*b–f*). The results displayed that the expression changes of these five genes is reasonable when the *BmSPR* gene mutates, which also confirms the conservation of $BH_4$ synthesis and metabolic pathways in silkworm and humans.

## 3.3. Normal development and negative locomotion activities in the silkworm *lemon* mutant

The clinical manifestations of SR deficiency include stunting, dyskinesia and dystonia. Therefore, we measured the body weight and length of the silkworm *lemon* mutant. The results showed that the

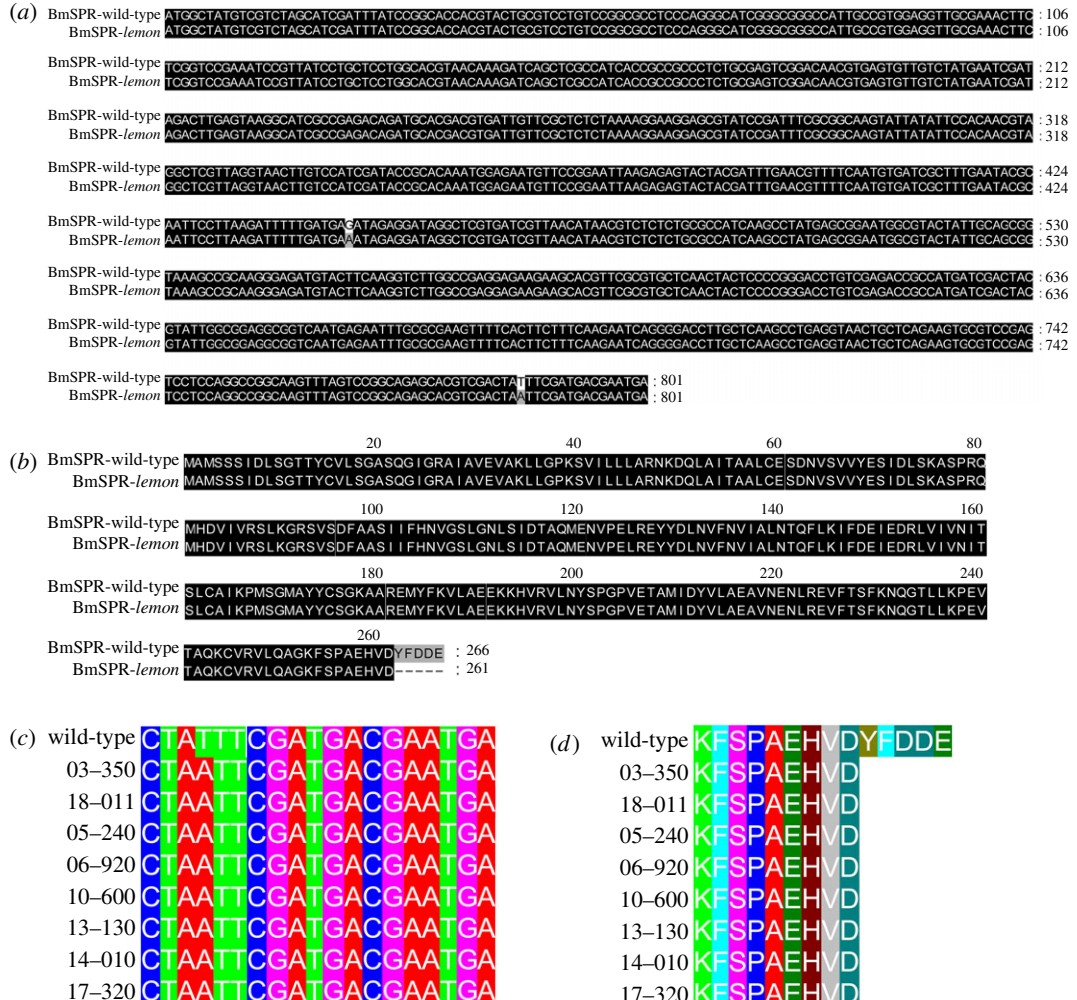

**Figure 3.** Alignment of the *BmSPR* gene sequence and amino acid sequence. (*a,b*) CDS and amino acid sequence alignment map of the *BmSPR* gene in the *lemon* mutant. (*c,d*) CDS and amino acid sequence alignment map of the *BmSPR* gene in the wild-type strain and eight yellow body colour silkworm mutants.

body length and weight of the *lemon* mutant and the wild-type silkworm were similar, meaning that the *BmSPR* mutation is not detrimental to body length and weight of the *lemon* mutant (figure 5*a,b*). We also measured the blood glucose level, but we did not investigate the difference between the silkworm *lemon* mutant and the wild-type (figure 5*c*). Then, we explored the locomotion ability of the *lemon* mutant through the mulberry leaf-luring test and the turnover test (figure 5*d,e*). We found that the *lemon* mutant needed more time to arrive the mulberry leaves than the control group; however, there was no significant difference in the time needed for the turnover test between the *lemon* mutant and the wild-type. According to these data, we believe that the *BmSPR* gene has a negative effect on the behavioural abilities of the *lemon* mutant.

## 3.4. Similar biochemical features of sepiapterin reductase deficiency in the *lemon* mutant

SR deficiency does not show hyperphenylalaninemia, which is one of its diagnostic features. Therefore, we compared the phenylalanine content in the blood of the *lemon* mutant and the wild-type. We found that the blood phenylalanine content was not significantly different between the *lemon* mutant and the wild-type (figure 6*a*). The dopamine and serotonin levels in the cerebrospinal fluid of the SR deficiency patients is significantly low, which is an important indicator for clinical diagnosis. We assessed the content of these two neurotransmitters in the brains of the *lemon* mutant and the wild-type by UPLC. We found that the levels of dopamine and serotonin in the *lemon* mutant were significantly lower than those in the wild-type (figure 6*b,c*). SR deficiency shows an increased level of

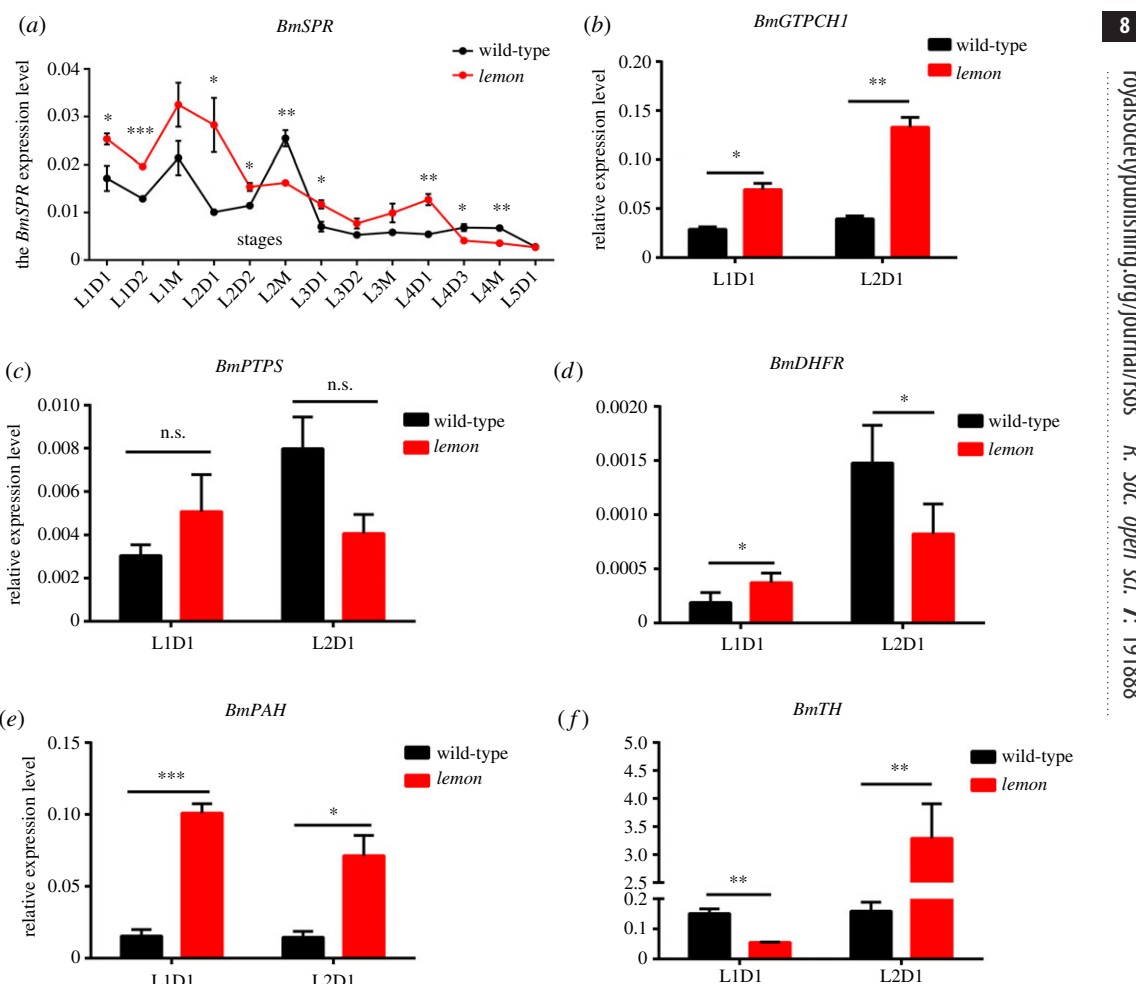

**Figure 4.** Expression of the *BmSPR* gene and five key genes of the BH$_4$ synthetic and metabolic pathways. (*a*) Temporal expression profile of *BmSPR* in the *lemon* mutant. (*b–f*) The relative expression level of key genes in the BH$_4$ synthetic and metabolic pathways. Student's *t*-test, $n = 3$, $^*p < 0.05$, $^{**}p < 0.01$, $^{***}p < 0.001$.

neopterin. We assayed the level of neopterin in the brains of the *lemon* mutant and the wild-type. We found that the content of neopterin in the *lemon* mutant was significantly higher than that in the wild-type (figure 6*d*). All data were consistent with the clinical features of SR deficiency; therefore, the *lemon* mutant meets the biochemical requirements as an animal model of SR deficiency.

## 3.5. Increased level of dopamine by oral supplementation of L-dopa in the *lemon* mutant

The main treatment for SR deficient patients is to correct the CNS dopamine deficiency by treatment with L-dopa and carbidopa. Therefore, we gave the *lemon* mutant three concentrations of these two compounds. The results showed that the dopamine level in the brains of silkworms in the three experimental groups was significantly higher than that in the control group, which is consistent with the treatment outcome of SR deficiency at the biochemical level (figure 7). However, there was no significant change in the phenotype, which might be because the concentrations of L-dopa and carbidopa were too low to form enough dopamine for melanin synthesis.

## 4. Discussion

SR deficiency is a monoamine neurotransmitter disorder caused by abnormalities in the synthesis, transport and metabolism of monoamines. SR deficiency is difficult to diagnose accurately and lacks effective medicines for a diverse patient population, which misses the best cure time and causes nerve injury. Thus, animal models of SR deficiency are required for further studies. Meng *et al.* [43] proved the Japanese *lem* mutant is caused by the mutation of the *BmSPR* gene. There is a mutant called *lemon*

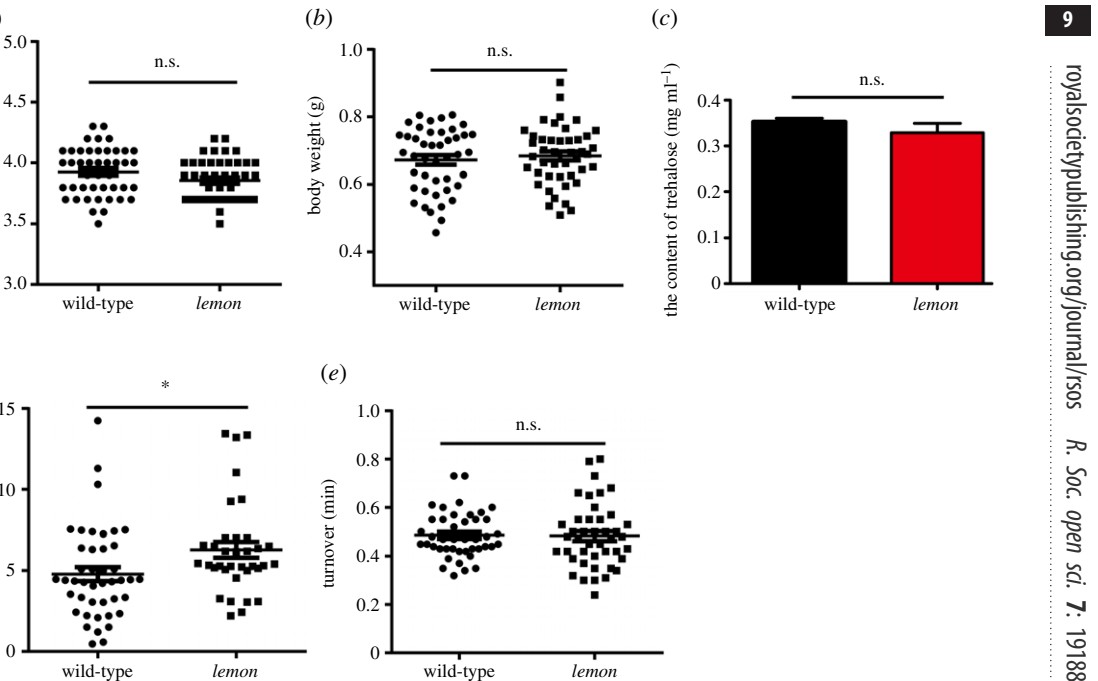

**Figure 5.** Normal development and locomotion activities in the *lemon* mutant. (*a,b*) A comparison of body length and weight between the *lemon* mutant and the wild-type, $n = 44$. (*c*) A comparison of the trehalose level between the *lemon* mutant and the wild-type, $n = 3$. (*d,e*) A comparison of locomotion between the *lemon* mutant and the wild-type, $n = 43$. Student's *t*-test, $^*p < 0.05$.

whose body colour is similar to the Japanese *lem*'s in the Southwest University. In this study, we aimed to explore whether the *lemon* mutant could serve as a model of SR deficiency. We found that the *lemon* mutant shows a *BmSPR* mutation, negative motor ability, normal phenylalanine level, decreased dopamine and serotonin levels, and increased neopterin level. These characteristics of the *lemon* mutant resemble the symptoms of SR deficiency in humans. Moreover, the administration of L-dopa and carbidopa increased the dopamine level of the *lemon* mutant. These findings support that the *lemon* mutant can be regarded as an animal model of SR deficiency in terms of genetic and biochemical aspects that are useful for studying precise diagnosis and for screening effective compounds.

*BmSPR* was highly expressed in the first and second instar larvae in the *lemon* mutant, suggesting that the gene is more important at an early stage in silkworms, which is consistent with the outcomes of SR deficiency in infancy in humans. In addition, the expression of other key genes in the BH$_4$ pathway suggests that it is conserved between silkworms and humans. However, we did not observe growth retardation, which is a classical clinical feature of SR deficient patients, in the *lemon* mutant. This might be because of species differences. The CDS of the human *SPR* gene is 786 bp, which encodes a 261 amino acid protein. Meanwhile, the CDS of the *BmSPR* gene is 801 bp, and the SPR enzyme is 266 amino acids long. Therefore, the function of *BmSPR* may differ between species. Administration of L-dopa is the main treatment for SR deficiency and aims to restore the dopamine level to rescue nerve damage. Although the dopamine level of the *lemon* mutant increased remarkably after feeding with L-dopa, we did not observe a significant phenotype change. In insects, dopamine is involved in the synthesis of melanin. Therefore, we suggest that the concentrations of L-dopa and carbidopa were too low to form enough dopamine for melanin synthesis, or the content of synthesized melanin was too low to show significant body colour changes.

The silkworm has many advantages and potential as an animal model of human diseases. With the completion of the genome map, as well as fine mapping and multi-strain re-sequencing for the silkworm, analysis of silkworm mutant mechanisms has improved. Rich silkworm mutant resources could potentially be used as disease models. The advantages of spontaneous animal models are that the occurrence and development of the diseases may be very similar to the corresponding diseases in humans. These silkworm mutants can not only help us understand the pathogenesis of various human diseases but also increase the possibility of discovering new genes for these diseases. There are 8469 human homologous genes in the silkworm with a 58% homology rate, which is close to the homology

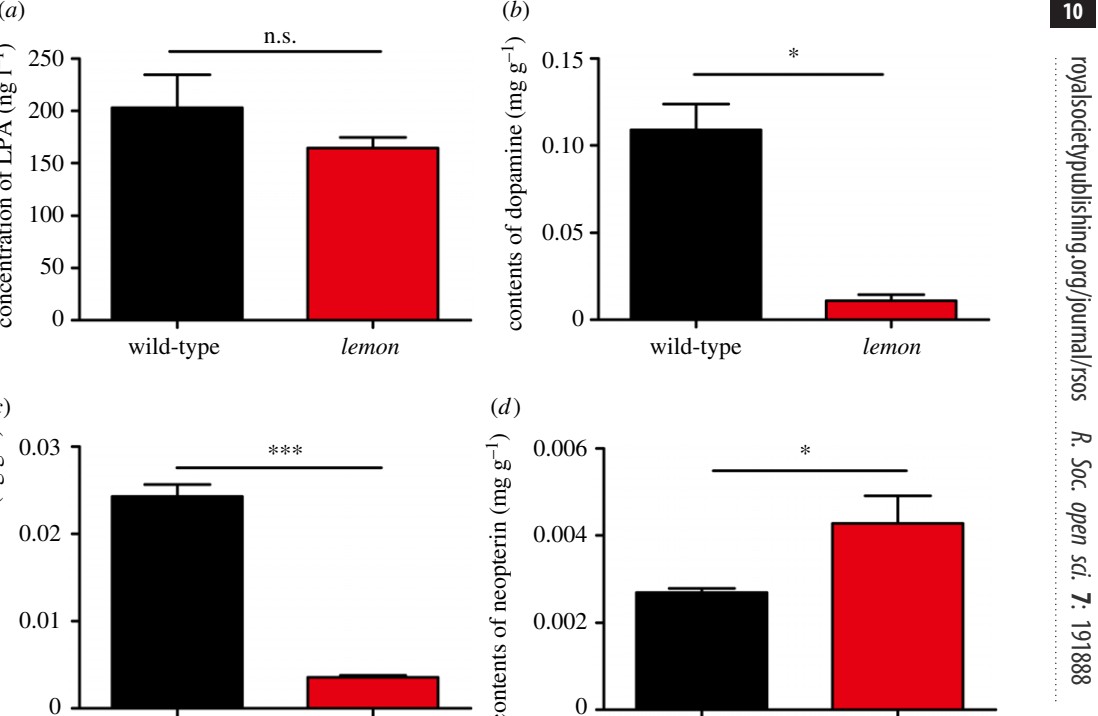

**Figure 6.** Similar biochemical features of SR deficiency in the silkworm *lemon* mutant. (*a*) A comparison of the LPA level between the *lemon* mutant and the wild-type. (*b,c*) A comparison of the dopamine and serotonin levels between the *lemon* mutant and the wild-type. (*d*) A comparison of the neopterin level between the *lemon* mutant and the wild-type. Student's *t*-test, $n = 3$, $^*p < 0.05$, $^{***}p < 0.001$.

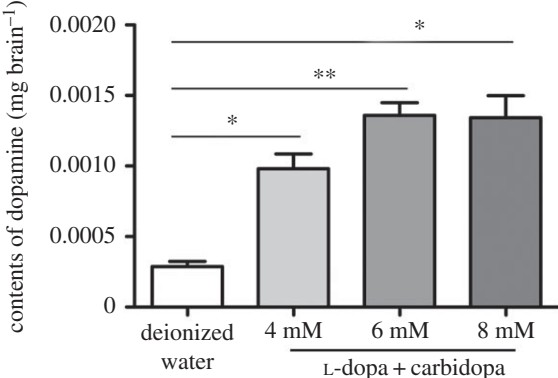

**Figure 7.** Increased dopamine level by oral supplementation of L-dopa and carbidopa in the silkworm *lemon* mutant. Student's *t*-test, $n = 3$, $^*p < 0.05$, $^{**}p < 0.01$.

rate of *Drosophila melanogaster*. There are 5006 genes related to 1612 human diseases which have corresponding homologous groups in the silkworm. These diseases are classified into 18 categories. The corresponding homologous genes in the silkworm are mainly related to skeletal, head, neck, neurological and developmental diseases. Therefore, the silkworm is a very promising animal model of human diseases, which can play a crucial part in revealing disease mechanisms and drug screening.

# 5. Conclusion

In summary, the *lemon* mutant is the first invertebrate model of human SR deficiency. We believe it will be an important resource to address questions of better diagnostic criteria and effective therapies for SR deficiency and other neurotransmitter disorders.

Data accessibility. The article's supporting data and relevant research materials such as statistical tools, protocols and software can be accessed. Our data are deposited at Dryad: https://doi.org/10.5061/dryad.gxd2547gr [45].

Authors' contributions. G.J.'s work included designing the study, carrying out the experiments, analysing the data and writing the manuscript. J.S. was responsible for the design of the study and the modification of the manuscript. H.H. reared silkworms and participated in the design of the study. X.T. designed the study. F.D. came up with the idea, designed the study and critically revised the manuscript. All authors agreed to publish the manuscript.

Competing interests. We declare we have no competing interests.

Funding. This work was supported by the National Natural Science Foundation of China (grant no. 31830094), the Hi-Tech Research and Development 863 Program of China (grant no. 2013AA102507), the Fundamental Research Funds for the Central Universities in China (grant no. XDJK2019C014), project funded by Chongqing Special Postdoctoral Science Foundation (grant no. XmT2018058) and Funds of China Agriculture Research System (grant no. CARS-18-ZJ0102).

Acknowledgements. We would like to thank the editor and the anonymous reviewers' valuable comments as well as the funding agencies for financial support for the development of this study.

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
