## [Reviewer comments · Royal Society Open Science]

Review History

RSOS-191888.R0 (Original submission)

Review form: Reviewer 1

Is the manuscript scientifically sound in its present form?

Yes

Are the interpretations and conclusions justified by the results?

Yes

Is the language acceptable?

Yes

Do you have any ethical concerns with this paper?

No

Have you any concerns about statistical analyses in this paper?

No

Recommendation?

Accept with minor revision (please list in comments)

Comments to the Author(s)

In this paper, the authors investigated an SPR orthologous gene mutant lemon in the silkworm and indicates that this silkworm lemon mutant could be a promising candidate model for SRD. The text is well arranged and the logic is clear. It is helpful for reader to understand the potential application value of this study because of the systematic results. Nonetheless, the manuscript needs to be reviewed by a native speaker. I recommend to you that this manuscript can be accepted after revision. The following are the questions and some mistakes in this manuscript:

1. A mouse model for human sepiapterin reductase deficiency (SR deficiency) has been reported so far. Please explain the reasons that you choose silkworm lemon mutant to construct the SRD model?
2. I don't understand that there is no body color change after feeding with L-dopa while you indicated that the yellow body color could be as the marker in your experiment?
3. Why do you perform the phylogenetic analysis of the SPR gene in these animals? What is the conclusion you want to show by phylogenetic analysis of the SPR gene?
4. Whether the SPR enzyme activity was affected by the BmSPR gene mutation?
5. The expression level of BmSPR gene in lemon mutant is higher than that in other periods. Why you suggest that the lemon mutant show similar characters to SR deficiency patients
6. Are these eight yellow-body color silkworms the same strain? Why do you think there is relationship between yellow-body color and BmSPR gene?
7. The homovanillic acid and 5-hydroxyindoleacetic acid are the important clinical indicators of SR deficiency. Why you solely detect dopamine and 5-hydroxytryptamine?
8. There is no difference in blood glucose level and development between lemon and the control group, which is not consistent with the SR deficiency patients' character. How to explain?
9. The units for dopamine and 5-hydroxytryptamine in Fig.7 and Fig.8 are different, please explain?
10. More deeply reason should be elucidated by using the mulberry leaf-luring and turnover test to explore the effect of BmSPR gene mutation on lemon mobility?

Review form: Reviewer 2

Is the manuscript scientifically sound in its present form?

Yes

Are the interpretations and conclusions justified by the results?

Yes

Is the language acceptable?

No

Do you have any ethical concerns with this paper?

No

Have you any concerns about statistical analyses in this paper?

No

Recommendation?

Accept with minor revision (please list in comments)

Comments to the Author(s)

This manuscript studied silkworm lemon mutant, and thought it could be used as the candidate animal model of sepiapterin reductase deficiency, They selected 5 key genes of BH4 synthetic and metabolic pathway, and thought lem was an important resource to address some

problems like better diagnostic criteria and effective therapies for SR deficiency and other neurotransmitter disorders. This research sound interesting, and I think it could be accepted after minor revision.

1. The language should be revised carefully.
2. Fig 1 should tell us the new findings.
3. P.16. Line 28, "In 2009, Meng demonstrated that BmSPR gene was responsible for the Japanese lem mutant".I can not find Meng's paper in the list of references.
- 4.Fig 6. The authors gave a model, but we can not know the exact distance the individuals for their locomotion activities.

Decision letter (RSOS-191888.R0)

12-Dec-2019

Dear Dr Dai,

The editors assigned to your paper ("The evaluation of silkworm lemon mutant as an invertebrate animal model for human sepiapterin reductase deficiency") have now received comments from reviewers. We would like you to revise your paper in accordance with the referee and Associate Editor suggestions which can be found below (not including confidential reports to the Editor). Please note this decision does not guarantee eventual acceptance.

Please submit a copy of your revised paper before 04-Jan-2020. Please note that the revision deadline will expire at 00.00am on this date. If we do not hear from you within this time then it will be assumed that the paper has been withdrawn. In exceptional circumstances, extensions may be possible if agreed with the Editorial Office in advance. We do not allow multiple rounds of revision so we urge you to make every effort to fully address all of the comments at this stage. If deemed necessary by the Editors, your manuscript will be sent back to one or more of the original reviewers for assessment. If the original reviewers are not available, we may invite new reviewers.

- Data accessibility

If you wish to submit your supporting data or code to Dryad (<http://datadryad.org/>), or modify your current submission to dryad, please use the following link:
<http://datadryad.org/submit?journalID=RSOS&manu=RSOS-191888>

- Competing interests

- Authors' contributions

- Acknowledgements

- Funding statement

Kind regards,
Anita Kristiansen
Editorial Coordinator
Royal Society Open Science
openscience@royalsociety.org

on behalf of Dr Robson da Costa (Associate Editor) and Kevin Padian (Subject Editor)
openscience@royalsociety.org

Subject Editor's comments (Professor Kevin Padian):

Thank you for submitting your manuscript. As you can see the reviewers are encouraging but they have some concerns that we hope you can address. Please submit a revised manuscript along with a marked-up version to track your changes. If you need more time than allotted for revision, just contact the editorial office. Most importantly, the quality of the English in the manuscript has been flagged as needing considerable improvement. If the resubmission is not in publishable English (and apologies for the great difficulty and irregularity of the English language), regrettably we will be unable to consider it further. Best wishes for your revision.

Associate Editor's comments (Dr Robson da Costa):

In the present study, Guihua Jiang and co-workers have characterised an insect model of human sepiapterin reductase deficiency, an inherited autosomal recessive disorder with unknown incidence. The authors have proposed the use of silkworm lemon mutant insect as a potential invertebrate model of sepiapterin reductase deficiency; they have found several biochemical changes and behavioural abnormalities in silkworm lemon mutant that are also found in patients. The proposed model could represent an important tool for the understanding of the pathological mechanisms involved in this syndrome and for the searching of new and effective compounds.

In addition to the reviewer's comments, the authors should particularly address the following criticisms:

- 1) The reading is sometimes difficult to follow and there are a few grammar mistakes. I strongly suggest the MS is carefully revised by a native English speaker.
- 2) Please, provide a reference/citation for the date Figure 1. If the figure is not original, please provide its source.
- 3) Please, improve the quality of the images in Fig. 2A and 2B. The background for wild-type and lemon silkworm should be the same.
- 4) The advantages and disadvantages of the present model should be discussed in comparison to the previously published mouse model for human sepiapterin reductase deficiency.

Reviewers' Comments to Author:

Reviewer: 1

Comments to the Author(s)

In this paper, the authors investigated an SPR orthologous gene mutant lemon in the silkworm and indicates that this silkworm lemon mutant could be a promising candidate model for SRD. The text is well arranged and the logic is clear. It is helpful for reader to understand the potential application value of this study because of the systematic results. Nonetheless, the manuscript needs to be reviewed by a native speaker. I recommend to you that this manuscript can be accepted after revision. The following are the questions and some mistakes in this manuscript:

1. A mouse model for human sepiapterin reductase deficiency (SR deficiency) has been reported so far. Please explain the reasons that you choose silkworm lemon mutant to construct the SRD model?
2. I don't understand that there is no body color change after feeding with L-dopa while you indicated that the yellow body color could be as the marker in your experiment?
3. Why do you perform the phylogenetic analysis of the SPR gene in these animals? What is the conclusion you want to show by phylogenetic analysis of the SPR gene?
4. Whether the SPR enzyme activity was affected by the BmSPR gene mutation?
5. The expression level of BmSPR gene in lemon mutant is higher than that in other periods. Why you suggest that the lemon mutant show similar characters to SR deficiency patients?
6. Are these eight yellow-body color silkworms the same strain? Why do you think there is relationship between yellow-body color and BmSPR gene?

7. The homovanillic acid and 5-hydroxyindoleacetic acid are the important clinical indicators of SR deficiency. Why you solely detect dopamine and 5-hydroxytryptamine?
8. There is no difference in blood glucose level and development between lemon and the control group, which is not consistent with the SR deficiency patients' character. How to explain?
9. The units for dopamine and 5-hydroxytryptamine in Fig.7 and Fig.8 are different, please explain?
10. More deeply reason should be elucidated by using the mulberry leaf-luring and turnover test to explore the effect of BmSPR gene mutation on lemon mobility?

Reviewer: 2

Comments to the Author(s)

This manuscript studied silkworm lemon mutant, and thought it could be used as the candidate animal model of sepiapterin reductase deficiency, They selected 5 key genes of BH4 synthetic and metabolic pathway, and thought lem was an important resource to address some problems like better diagnostic criteria and effective therapies for SR deficiency and other neurotransmitter disorders. This research sound interesting, and I think it could be accepted after minor revision.

1. The language should be revised carefully.
2. Fig 1 should tell us the new findings.
3. P.16. Line 28, "In 2009, Meng demonstrated that BmSPR gene was responsible for the Japanese lem mutant".I can not find Meng's paper in the list of references.
- 4.Fig 6. The authors gave a model, but we can not know the exact distance the individuals for their locomotion activities.

Author's Response to Decision Letter for (RSOS-191888.R0)

See Appendices A & B.

RSOS-191888.R1 (Revision)

Review form: Reviewer 2

Is the manuscript scientifically sound in its present form?

Yes

Are the interpretations and conclusions justified by the results?

Yes

Is the language acceptable?

Yes

Do you have any ethical concerns with this paper?

No

Have you any concerns about statistical analyses in this paper?

No

Recommendation?

Accept as is

Comments to the Author(s)

This version I think can be accepted. I only wonder the staff gauge in Fig 2A right? 1mm? or 1cm? the difference between 2A and 2B so big!

Review form: Reviewer 3 (Hiroshi Hamamoto)**Is the manuscript scientifically sound in its present form?**

Yes

Are the interpretations and conclusions justified by the results?

Yes

Is the language acceptable?

Yes

Do you have any ethical concerns with this paper?

No

Have you any concerns about statistical analyses in this paper?

No

Recommendation?

Accept with minor revision (please list in comments)

Comments to the Author(s)

In this manuscript, the authors argue that the lemon mutant is a useful model to develop diagnostic criteria and screen novel therapies for SR deficiency. The authors replied almost adequately to the previous reviewer's raised points. However, I think it required further correction for a few issues.

1. The authors need to have respect attitude to previous work, which identified responsible gene for Japanese lemon mutant, Meng Y et al. The authors need to mention, 1) mutation point of this lemon mutant is same as BmmtSPR, 2) what are advance from previous work, 3) Figure 3; body color of lemon mutant was same as previous work.

2. Page 19, line 4-5; page 22, 19-22; The authors suggested that the SPR gene was relatively conserved from the phylogenetic tree, while the function of BmSPR may differ since the length of the gene is different. These two explanation has a discrepancy. I think the results of phylogenetic tree analysis suggested the BmSPR is a little far from human SPR. I recommend the authors omit this figure and section because this analysis did not make sense regarding functional conservation.

3. Figure 8; how to administrate L-dopa+carbidopa to silkworm? Is it possible to estimate the ingestion dose of these drugs?

Decision letter (RSOS-191888.R1)

14-Feb-2020

Dear Dr Dai,

On behalf of the Editors, I am pleased to inform you that your Manuscript RSOS-191888.R1 entitled "Evaluation of the silkworm lemon mutant as an invertebrate animal model for human sepiapterin reductase deficiency" has been accepted for publication in Royal Society Open Science subject to minor revision in accordance with the referee suggestions. Please find the referees' comments at the end of this email.

The reviewers and Subject Editor have recommended publication, but also suggest some minor revisions to your manuscript. Therefore, I invite you to respond to the comments and revise your manuscript.

- Ethics statement

- Data accessibility

If you wish to submit your supporting data or code to Dryad (<http://datadryad.org/>), or modify your current submission to dryad, please use the following link:
<http://datadryad.org/submit?journalID=RSOS&manu=RSOS-191888.R1>

- Competing interests

- Authors' contributions

AB carried out the molecular lab work, participated in data analysis, carried out sequence alignments, participated in the design of the study and drafted the manuscript; CD carried out the statistical analyses; EF collected field data; GH conceived of the study, designed the study,

coordinated the study and helped draft the manuscript. All authors gave final approval for publication.

- Acknowledgements

- Funding statement

Because the schedule for publication is very tight, it is a condition of publication that you submit the revised version of your manuscript before 23-Feb-2020. Please note that the revision deadline will expire at 00.00am on this date. If you do not think you will be able to meet this date please let me know immediately.

on behalf of Dr Robson da Costa (Associate Editor) and Kevin Padian (Subject Editor)
openscience@royalsociety.org

Associate Editor Comments to Author (Dr Robson da Costa):

The authors have addressed most of the comments raised by both reviewers. However, there are few points that remain to be addressed as pointed by both reviewers.

Reviewer comments to Author:
Reviewer: 2
Comments to the Author(s)

This version I think can be accepted. I only wonder the staff gauge in Fig 2A right? 1mm? or 1cm? the difference between 2A and 2B so big!

Reviewer: 3
Comments to the Author(s)

In this manuscript, the authors argue that the lemon mutant is a useful model to develop diagnostic criteria and screen novel therapies for SR deficiency. The authors replied almost adequately to the previous reviewer's raised points. However, I think it required further correction for a few issues.

1. The authors need to have respect attitude to previous work, which identified responsible gene for Japanese lemon mutant, Meng Y et al. The authors need to mention, 1) mutation point of this lemon mutant is same as BmmtSPR, 2) what are advance from previous work, 3) Figure 3; body color of lemon mutant was same as previous work.
2. Page 19, line 4-5; page 22, 19-22; The authors suggested that the SPR gene was relatively conserved from the phylogenetic tree, while the function of BmSPR may differ since the length of the gene is different. These two explanation has a discrepancy. I think the results of phylogenetic tree analysis suggested the BmSPR is a little far from human SPR. I recommend the authors omit this figure and section because this analysis did not make sense regarding functional conservation.
3. Figure 8; how to administrate L-dopa+carbidopa to silkworm? Is it possible to estimate the ingestion dose of these drugs?

Author's Response to Decision Letter for (RSOS-191888.R1)

See Appendix C.

Decision letter (RSOS-191888.R2)

27-Feb-2020

Dear Dr Dai,

It is a pleasure to accept your manuscript entitled "Evaluation of the silkworm lemon mutant as an invertebrate animal model for human sepiapterin reductase deficiency" in its current form for publication in Royal Society Open Science. The comments of the reviewer(s) who reviewed your manuscript are included at the foot of this letter.

on behalf of Dr Robson da Costa (Associate Editor) and Kevin Padian (Subject Editor)
openscience@royalsociety.org

Appendix A

upload a file "Response to Referees" in "Section 6 - File Upload"

Subject Editor's comments (Professor Kevin Padian):

Thank you for submitting your manuscript. As you can see the reviewers are encouraging but they have some concerns that we hope you can address. Please submit a **revised manuscript** along with a **marked-up version** to track your changes. If you need more time than allotted for revision, just contact the editorial office. Most importantly, the quality of the English in the manuscript has been flagged as needing considerable improvement. If the resubmission is not in publishable English (and apologies for the great difficulty and irregularity of the English language), regrettably we will be unable to consider it further. Best wishes for your revision.

Associate Editor's comments (Dr Robson da Costa):

In the present study, Guihua Jiang and co-workers have characterised an insect model of human sepiapterin reductase deficiency, an inherited autosomal recessive disorder with unknown incidence. The authors have proposed the use of silkworm lemon mutant insect as a potential invertebrate model of sepiapterin reductase deficiency; they have found several biochemical changes and behavioural abnormalities in silkworm lemon mutant that are also found in patients. The proposed model could represent an important tool for the understanding of the pathological mechanisms involved in this syndrome and for the searching of new and effective compounds.

Dear editors,

Thank you for sending our manuscript for review and for the valuable comments given by the reviewers, allowing us the opportunity to improve the paper in depth. Here, we submit a revised version of our manuscript entitled "Evaluation of the silkworm lemon mutant as an invertebrate animal model for human sepiapterin reductase deficiency" (ID: RSOS-191888), which has been modified according to the editor's and reviewers' suggestions, with point by point responses given below. Efforts were also made to correct imprecisions and add crucial details. We marked all changes in **red** in the revised manuscript. Thank you again for your attention and consideration.

Sincerely,

Fang-yin DAI

In addition to the reviewer's comments, the authors should particularly address the following criticisms:

1) The reading is sometimes difficult to follow and there are a few grammar mistakes. I strongly suggest the MS is carefully revised by a native English speaker.

Response: Thank you. We corrected the mistakes and improved the English with help from a professional language editorial service company with high-level native English speakers.

2) Please, provide a reference/citation for the date Figure 1. If the figure is not original, please provide its source.

Response: I have added the reference "[22] Thöny B, Auerbach G, Blau N. Tetrahydrobiopterin biosynthesis, regeneration and functions [J]. Biochemical Journal, 2000, 347 Pt 1: 1-16" to the list of references.

3) Please, improve the quality of the images in Fig. 2A and 2B. The background for wild-type and lemon silkworm should be the same.

Response: Thank you for the valuable suggestion. We have improved the quality of the images in Fig. 2A and 2B.

4) The advantages and disadvantages of the present model should be discussed in comparison to the previously published mouse model for human sepiapterin reductase deficiency.

Response: Although a murine model for human SR deficiency has been reported in 2008, the *lemon* mutant has some advantages in comparison to it. First, the *lemon* mutant is easy to obtain. The *lemon* mutant is a stably inherited mutant preserved in the Silkworm Gene Bank of Southwest University. However, the murine SR deficiency model was obtained by gene knockout, which required a long experimental time and complex steps. Second, the *lemon* mutant may better reflect SR deficiency. The *lemon* mutant is a spontaneous animal model that better simulates the occurrence and development of human disease under natural conditions. The murine SR deficiency model was constructed by gene knockout, so the occurrence and formation of SR deficiency may be different from it forming naturally. Third, the blood phenylalanine content in the *lemon* mutant is normal, which is consistent with the clinical manifestation in SR deficient patients. However, the level of blood phenylalanine in the murine SR deficiency model is increased.

The *lemon* mutant still has disadvantages as an SR deficiency model. For example, the locomotor ability of silkworms is not as strong as that of mice, so the effect of an *SPR* gene mutation on the locomotor ability in the *lemon* mutant may not be easily detected or not be significantly displayed.

Reviewers' Comments to Author:

Reviewer: 1

Comments to the Author(s)

In this paper, the authors investigated an SPR orthologous gene mutant lemon in the silkworm and indicates that this silkworm lemon mutant could be a promising candidate model for SRD. The text is well arranged and the logic is clear. It is helpful for reader to understand the potential application value of this study because of the systematic results. Nonetheless, the manuscript needs to be reviewed by a native speaker. I recommend to you that this manuscript can be accepted after revision. The following are the questions and some mistakes in this manuscript:

1. A mouse model for human sepiapterin reductase deficiency (SR deficiency) has been reported so far. Please explain the reasons that you choose silkworm lemon mutant to construct the SRD model?

Response: As an SR deficiency model, the *lemon* mutant has some advantages in comparison to mice. For example, the experimental cycle is much shorter. We can obtain the *lemon* mutant easily from the Silkworm Gene Bank at Southwest University. On the other hand, the mouse SR deficiency model was constructed by gene knockout, as noted above. Moreover, the *lemon* mutant showed a normal blood phenylalanine content like human SR deficient patients, but the mouse SR deficiency model did not.

2. I don't understand that there is no body color change after feeding with L-dopa while you

indicated that the yellow body color could be as the marker in your experiment?

Response: Presumably, this is because the concentration of levodopa and carbidopa was too low to synthesize melanin, or the synthetic melanin content was not enough to change the body color clearly.

3. Why do you perform the phylogenetic analysis of the SPR gene in these animals? What is the conclusion you want to show by phylogenetic analysis of the SPR gene?

Response: To understand the conservation of the *SPR* gene between human and common model animals, we performed a phylogenetic analysis of the *SPR* gene. We found that the *SPR* gene is relatively conserved in different evolutionary classes, which may be helpful for finding another animal model of SR deficiency.

4. Whether the SPR enzyme activity was affected by the BmSPR gene mutation?

Response: We analyzed the SPR protein domain, and found that the amino acid missing from the SPR enzyme of *lemon* mutant is in this domain, so we speculated that the enzyme activity of SPR was partially affected.

5. The expression level of BmSPR gene in lemon mutant is higher than that in other periods. Why do you suggest that the lemon mutant show similar characters to SR deficiency patients

Response: The expression level of the *BmSPR* gene in the *lemon* mutant is higher in early larva, indicating that it is more important at that time. SR deficient patients show clinical manifestations in infancy, which means the *SPR* gene is very important in infancy. So, we suggest that the *lemon* mutant shows similar characters to SR deficiency patients.

6. Are these eight yellow-body color silkworms the same strain? Why do you think there is relationship between yellow-body color and BmSPR gene?

Response: The eight yellow-body color strains have not been shown in the same system until we found that the *BmSPR* gene has the same mutation. Including the *lemon* mutant, the nine mutants had the same body color and *BmSPR* mutation, so we believe that there is a relationship between the yellow-body color and the *BmSPR* gene.

7. The homovanillic acid and 5-hydroxyindoleacetic acid are the important clinical indicators of SR deficiency. Why you solely detect dopamine and 5-hydroxytryptamine?

Response: First, dopamine and 5-hydroxytryptamine are detected in SR deficient patients clinically. Second, homovanillic acid and 5-hydroxyindoleacetic acid are downstream metabolites of dopamine and 5-hydroxytryptamine, respectively, in humans, which are different in silkworms. Therefore, we detected only dopamine and 5-hydroxytryptamine.

8. There is no difference in blood glucose level and development between lemon and the control group, which is not consistent with the SR deficiency patients' character. How to explain?

Response: Blood glucose level and organism development are affected by many factors, especially in different species. Therefore, we speculate that this may be the result of species differences. In other words, the function of the *SPR* gene may differ partly in different animals.

9. The units for dopamine and 5-hydroxytryptamine in Fig.7 and Fig.8 are different, please explain?

Response: The detections of dopamine and 5-hydroxytryptamine in Figure 7 need to be qualitative and quantitative, so we weighed the heads and found that the *lemon* head was heavier than the wild type. The levels of dopamine and 5-hydroxytryptamine are lower in the *lemon* mutant, so we think it was not necessary to weigh their heads in the feeding test. Thus, the units used in the two figures are different.

10. More deeply reason should be elucidated by using the mulberry leaf-luring and turnover test to explore the effect of BmSPR gene mutation on lemon mobility?

Response: SR deficient patients show dystonia, which can affect their motor ability. Both the mulberry leaf-luring test and the turnover test require muscle strength for the *lemon* mutant. That is why we designed these two tests.

Reviewer: 2

Comments to the Author(s)

This manuscript studied silkworm lemon mutant, and thought it could be used as the candidate animal model of sepiapterin reductase deficiency, they selected 5 key genes of BH4 synthetic and metabolic pathway, and thought *lem* was an important resource to address some problems like better diagnostic criteria and effective therapies for SR deficiency and other neurotransmitter disorders. This research sound interesting, and I think it could be accepted after minor revision.

1. The language should be revised carefully.

Response: Thank you. We corrected the mistakes and improved the English with the help of a professional language editorial service company with high-level native English experts.

2. Fig 1 should tell us the new findings.

Response: Fig. 1 aims to show the three biosynthetic pathways of BH4 and to introduce the SPR functions in this pathway. Thöny *et al.* reported the pathway in 2000 (Thöny B, Auerbach G, Blau N. Tetrahydrobiopterin biosynthesis, regeneration and functions [J]. Biochemical Journal, 2000, 347 Pt 1: 1-16). There are no new findings for this pathway in this manuscript.

3. P.16. Line 28, “In 2009, Meng demonstrated that BmSPR gene was responsible for the Japanese *lem* mutant”. I can not find Meng’s paper in the list of references.

Response: I have added the reference “[44] Meng Y, Katsuma S, Daimon T, et al. The silkworm mutant *lemon* (*lemon lethal*) is a potential insect model for human sepiapterin reductase deficiency [J]. J Biol Chem, 2009, 284 (17): 11698-705. doi: 10.1074/jbc.M900485200.” to the list of references.

4. Fig 6. The authors gave a model, but we can not know the exact distance the individuals for their locomotion activities.

Response: In the mulberry leaf-luring test, silkworm *lemon* mutants and the wild-type were placed 0.5 cm away from the chronograph line, which is the pre-crawl distance. At 5 cm away from the chronograph line, we put mulberry leaves to induce silkworm crawling. When the

silkworms climbed to the timing line, we began to count. The number of silkworms crawling to the mulberry leaves and the time required by each group were counted within 15 minutes. Thus, the exact distance of the individuals was 5 cm.

Appendix B

International Science Editing

www.internationalscienceediting.com

DATE: December 31, 2019

Compuscript Ltd
T/A International Science Editing
Bay K, Shannon Industrial Park West
Shannon, Co Clare
Ireland
Phone +353 61 472818 Fax +353 61 472688

To whom it may concern,

The paper "Evaluation of the silkworm lemon mutant as an invertebrate animal model for human sepiapterin reductase deficiency" by Jiangbo Song was edited by International Science Editing. We were asked not to edit the references. Please contact us if you would like to view the edited paper.

Kindest regards,

David Cushley.

Appendix C

International Science Editing

www.internationalscienceediting.com

DATE: December 31, 2019

Compuscript Ltd
T/A International Science Editing
Bay K, Shannon Industrial Park West
Shannon, Co Clare
Ireland
Phone +353 61 472818 Fax +353 61 472688

To whom it may concern,

The paper "Evaluation of the silkworm lemon mutant as an invertebrate animal model for human sepiapterin reductase deficiency" by Jiangbo Song was edited by International Science Editing. We were asked not to edit the references. Please contact us if you would like to view the edited paper.

Kindest regards,

David Cushley.